# Role of Nucleobindin-2 in the Clinical Pathogenesis and Treatment Resistance of Glioblastoma

**DOI:** 10.3390/cells12192420

**Published:** 2023-10-09

**Authors:** I-Cheng Lin, Chih-Hui Chang, Yoon Bin Chong, Shih-Hsun Kuo, Yu-Wen Cheng, Ann-Shung Lieu, Tzu-Ting Tseng, Chien-Ju Lin, Hung-Pei Tsai, Aij-Lie Kwan

**Affiliations:** 1Department of Surgery, Kaohsiung Municipal Siaogang Hospital, Kaohsiung 81267, Taiwan; 1000456@kmuh.org.tw; 2Division of Neurosurgery, Department of Surgery, Kaohsiung Medical University Hospital, Kaohsiung 80708, Taiwan; chchang20@gmail.com (C.-H.C.); bin99068@hotmail.com (Y.B.C.); e791125@gmail.com (A.-S.L.); cawaii7992@gmail.com (T.-T.T.); 3Department of Radiation Oncology, Kaohsiung Medical University Hospital, Kaohsiung 80708, Taiwan; 1020001@kmuh.org.tw; 4Gradate Institute of Medicine, College of Medicine, Kaohsiung Medical University, Kaohsiung 80708, Taiwan; murraycheng1015@gmail.com; 5Department of Neurosurgery, Kaohsiung Veterans General Hospital, Kaohsiung 81362, Taiwan; 6Department of Surgery, School of Medicine, College of Medicine, Kaohsiung Medical University, Kaohsiung 80756, Taiwan; 7School of Pharmacy, College of Pharmacy, Kaohsiung Medical University, Kaohsiung 80708, Taiwan; mistylin@kmu.edu.tw; 8Department of Neurosurgery, University of Virginia, Charlottesville, VA 22904, USA

**Keywords:** glioblastoma, Nucleobindin-2, chemotherapy, radiotherapy

## Abstract

Glioblastoma (GBM) stands as the most prevalent primary malignant brain tumor, typically resulting in a median survival period of approximately thirteen to fifteen months after undergoing surgery, chemotherapy, and radiotherapy. Nucleobindin-2 (NUCB2) is a protein involved in appetite regulation and energy homeostasis. In this study, we assessed the impact of NUCB2 expression on tumor progression and prognosis of GBM. We further evaluated the relationship between NUCB2 expression and the sensitivity to chemotherapy and radiotherapy in GBM cells. Additionally, we compared the survival of mice intracranially implanted with GBM cells. High NUCB2 expression was associated with poor prognosis in patients with GBM. Knockdown of NUCB2 reduced cell viability, migration ability, and invasion ability of GBM cells. Overexpression of NUCB2 resulted in reduced apoptosis following temozolomide treatment and increased levels of DNA damage repair proteins after radiotherapy. Furthermore, mice intracranially implanted with NUCB2 knockdown GBM cells exhibited longer survival compared to the control group. NUCB2 may serve as a prognostic biomarker for poor outcomes in patients with GBM. Additionally, NUCB2 not only contributes to tumor progression but also influences the sensitivity of GBM cells to chemotherapy and radiotherapy. Therefore, targeting NUCB2 protein expression may represent a novel therapeutic approach for the treatment of GBM.

## 1. Introduction

Glioblastoma (GBM), an aggressive malignant tumor within the central nervous system, represents around 15% of all brain tumors and constitutes 54% of gliomas [1]. One of the challenges in treating GBM is its aggressive infiltration into normal brain tissue, making surgical removal difficult, especially in eloquent brain areas. The standard treatment for GBM involves surgery followed by concurrent chemotherapy and radiotherapy [2]. Temozolomide (TMZ), a DNA alkylating agent, is commonly used in the treatment of newly diagnosed GBM and has shown efficacy, increasing median overall survival by 2 months [3]. However, the median overall survival for GBM remains approximately 13–15 months, with a 2-year survival rate of only 8–12% [4]. The poor prognosis is attributed to the high resistance of GBM to TMZ and radiotherapy [5,6]. Ongoing studies are focused on targeting the DNA damage repair mechanism and developing radiosensitizers to improve GBM treatment.

Nucleobindin-2 (NUCB2) is a protein comprising 396 amino acids and possesses multiple functional domains, including a signal peptide, a Leu/Ile-rich region, two EF-hand domains, and a leucine zipper motif [7]. NUCB2 exhibits DNA/Ca^2+^ binding capabilities and can undergo cleavage by prohormone convertase, resulting in the formation of nesfatin-1, nesfatin-2, and nesfatin-3. In 2006, Oh-I et al. initially characterized nesfatin-1 as an anorexigenic factor in rats, while the roles of the other two fragments remain undiscovered [8]. NUCB2 is broadly distributed in both the central nervous system and peripheral tissues [9]. In peripheral tissues, NUCB2 can be found in adipose tissue, the pancreas, cardiomyocytes, and reproductive systems [10]. The physiological functions of nesfatin-1 include decreasing food intake, reducing gastrointestinal motility, and increasing insulin secretion [11,12,13]. NUCB2 also demonstrates antioxidant, anti-inflammatory, and anti-apoptotic properties in various disease contexts [14]. In addition to its involvement in energy regulation, NUCB2 has been identified as a negative prognostic indicator in several cancer types, including breast cancer, colon cancer, bladder cancer, prostate cancer, endometrial cancer, gastric cancer, papillary thyroid cancer, and renal cell carcinoma [15,16,17]. NUCB2 is implicated in various stages of cancer progression, where it facilitates tumorigenesis, invasion, migration, and epithelial-mesenchymal transition (EMT) of cancer cells [18,19,20,21]. Interestingly, NUCB2 plays a contrasting role in human adrenocortical and ovarian epithelial carcinoma by inhibiting tumor proliferation and promoting apoptosis [22,23]. A prior study conducted by Liu et al. reported that NUCB2 is overexpressed in GBM and that its elevated expression contributes to the growth and invasion of GBM cells. Moreover, high NUCB2 expression is associated with GBM recurrence [24].

The identification of novel therapeutic targets plays a pivotal role in advancing the treatment of GBM. The primary aim of this study is to examine the clinicopathological significance of NUCB2 in the progression of GBM and to delve into the connection between NUCB2 expression and the development of treatment resistance.

## 2. Materials and Methods

### 2.1. Patients

The participants in this research were chosen from the patient pool at the Division of Neurosurgery, Kaohsiung Medical University Hospital. Eligibility criteria for inclusion encompassed patients with glioma who possessed comprehensive medical records, complete follow-up data, robust pathological findings, and well-executed immunohistochemical staining. Conversely, individuals who were exclusively diagnosed through biopsy, had incomplete medical records, lacked follow-up data, exhibited subpar pathological results, or displayed inadequate immunohistochemical staining were excluded from the study.

### 2.2. Immunohistochemical Staining

Tissue sections measuring 3 mm in thickness were obtained from formalin-fixed, paraffin-embedded tissue samples collected from each patient. These sections underwent deparaffinization, rehydration, and antigen retrieval through autoclaving at 121 °C for 10 min in Target Retrieval solution (pH 6.0; S2369, Dako, Glostrup, Denmark). After a 20 min incubation at room temperature, endogenous peroxidase activity was quenched by the application of 3% hydrogen peroxide for 5 min at room temperature. Following two rinses with Tris buffer, the sections were incubated with a primary antibody at room temperature for 1 h. After an additional two rinses with Tris buffer, the sections were exposed to a secondary horseradish peroxidase-conjugated antibody at room temperature for 30 min. Subsequently, the slides were treated with 3,3-diaminobenzidine (K5007, Dako, Glostrup, Denmark) for 5 min, counterstained with Mayer’s hematoxylin for 90 s, and finally mounted with malinol.

Immunohistochemical staining results were categorized into low-level expression or high-level expression. Scores were assigned based on the proportion of tumor cells that were positively stained, with the following criteria: 0 (no positively stained tumor cells), 1 (<10% positive cells), 2 (10–50% positive cells), and 3 (>50% positive cells). Staining intensity was graded as 0 (no staining), 1 (weak staining), 2 (moderate staining), or 3 (strong staining). The staining index (SI) was calculated by multiplying the intensity score by the percentage of positively stained tumor cells, resulting in potential scores of 0, 1, 2, 3, 4, 6, and 9. A total score of 4 was set as the threshold, where scores > 4 were deemed indicative of high expression and scores < 3 were classified as low expression.

### 2.3. Cell Culture and NUCB2 siRNA Transfection

All cell lines were sourced from the Bioresource Collection and Research Center and were cultured at 37 °C in a 5% CO_2_ environment. GBM8401 cells were cultured in RPMI medium supplemented with 10% fetal bovine serum (FBS). U87-MG and SVGp12 cells were cultured in modified Eagle’s medium (MEM) supplemented with 10% FBS. G5T/VGH, Hs683, and A172 cells were cultured in Dulbecco’s MEM medium supplemented with 10% FBS. GBM8401, U87-MG, G5T/VGH, M059K, Hs683, and A172 cells were derived from GBM patients, while SVGp12 cells originated from normal tissue and were utilized as a control. For siRNA transfection of glioma cells, DharmaFECTTM Transfection Reagents (Dharmacon, Lafayette, CO, USA) were employed along with human NUCB2 siRNA constructs (Sigma, St. Louis, MO, USA) containing the following sequences: NUCB2 siRNA #1 sense strand: GGAUUCCCUUCAAGAUAUA, antisense strand: UAUAUCUUGAAGGGAAUCCA. A concentration of 5 μM of NUCB2 siRNA was used for transfection. Following siRNA transfection, cells were cultured for 3 days before further experimentation. NUCB2 protein levels were assessed using Western blot analysis.

### 2.4. Proliferation Assay

Cells were resuspended in culture medium supplemented with 10% FBS and were plated into individual wells of a 6-well culture dish at a density of approximately 1 million cells per well, with each well containing 2 milliliters of medium. Subsequently, the cells were incubated at the aforementioned conditions for time intervals of 24, 48, and 72 h. Following siRNA treatment, cell viability was determined using the 3-(4,5-dimethylthiazol-2-yl)-2,5-diphenyltetrazolium bromide (MTT) assay.

### 2.5. Migration Assay

Cell migration was assessed using a wound healing assay (80209; ibidi GmbH, Martinsried, Germany). A culture-insert was positioned in a 6-well plate and incubated at 37 °C for 12 h. Cells (70 μL) were seeded at a density of 1 × 10^5^ cells/mL and incubated for 24 h before siRNA transfection. Wound closure was observed and photographed at 24, 48, and 72 h following siRNA treatment. 

### 2.6. Invasion Assay

In vitro cell invasion assays were conducted using Transwell chambers (COR3452; CORNING, Corning, NY, USA). Cells were seeded at a density of 5 × 10^5^ cells per insert, and the lower chamber of each Transwell was filled with 2 mL of medium containing either nonsense siRNA (negative control) or NUCB2 siRNA. Following a 24 h incubation, cells that remained on the upper surfaces of the Transwell membranes were removed using cotton swabs. Cells that had successfully invaded through the membranes to the bottom of the insert were fixed, stained, photographed, and quantified by counting the number of cells in six randomly selected high-powered fields.

### 2.7. Western Blotting

All samples were lysed with 200 μL of lysis buffer. Subsequently, 50 μg of protein from each sample was loaded into the wells of a sodium dodecyl sulfate-polyacrylamide gel and subjected to electrophoresis at 50 V for 4 h. Following electrophoresis, the separated proteins were transferred onto poly (vinylidene fluoride) membranes. After a 1 h incubation in blocking buffer, the membranes were then exposed to primary antibodies (NUCB2 (PAB27403; 1:500; Abnova; Taipe, Taiwan), β-actin (A5441; 1:20,000; Sigma; USA), cyclin D1 (60186-1-lg; 1:500; Proteintech; Chicago, IL, USA), E-cadherin (20874-1-AP; 1:500; Proteintech; USA), N-cadherin (22018-1-AP; 1:500; Proteintech; USA), vascular endothelial growth factor (VEGF) (AP6290b; 1:500; ABGENT; San Diego, CA, USA), ku70 (sc-17789; 1:500; SANTA CRUZ; Santa Cruz, CA, USA, Europe), ku80 (ARG57867; 1:1000; Arigo; Hsinchu, Taiwan), Rad51 (GTX70230; 1:2000; GeneTex; Irvine, CA, USA), Rad52 (sc-365341; 1:500; SANTA CRUZ; Europe), cleaved caspase-3 (#9664; 1:500; Cell Signaling; Beverly, MA, USA), and poly (ADP-ribose) polymerases (PARP) (#9542; 1:500; Cell Signaling; USA)) for 16 h at 4 °C. The membranes were subsequently incubated with secondary antibodies, specifically goat anti-rabbit (AP132P, 1:5000; Millipore, Billerica, MA, USA) and goat anti-mouse (AP124P, 1:5000; Millipore), for a duration of 90 min. Specific protein bands were detected using an enhanced chemiluminescence solution (Western Lightning, 205-14621; Perkin Elmer, Waltham, MA, USA) and a MiniChemiTM imaging and analysis system (Beijing Sage Creation, Beijing, China).

### 2.8. Colony Formation Assay

GBM cells were seeded in 6-well plates at a density of 400 cells for the radiation doses of 2. A linear accelerator was used to irradiate cells at room temperature. After a 10-day incubation, the plates were stained with 0.5% crystal violet.

### 2.9. Animal Model

Athymic Balb/c mice (LASCO BioSciences) were used as the animal model. GBM cells with knock-down NUCB2 (1 × 10^4^ cells in a volume of 5 μL) were implanted intracranially in the striatum of the mice. Tumor size was observed through an in vivo imaging system (IVIS).

### 2.10. Data Analyses

Statistical analysis was performed using SPSS 19.0 software (SPSS, Inc., Chicago, IL, USA). The chi-squared test was employed to ascertain associations between NUCB2 protein expression and specific clinicopathologic parameters. Survival rate analyses were carried out using the Kaplan–Meier method and assessed using the log-rank test. Multivariate Cox regression analyses were performed to confirm the independent impact of each variable assessed in this study. To compare the results of the proliferation, migration, and invasion assays, one-way analysis of variance was utilized. A *p*-value of <0.05 was considered statistically significant for all analyses.

## 3. Results

### 3.1. Higher NUCB2 Expression Is Correlated with Poor Prognosis in GBM

A total of 99 patients diagnosed with glioma were included in the study. Based on immunohistochemical staining, patients were divided into low or high NUCB2 expression groups (Figure 1A). Among them, 34 patients exhibited low-level NUCB2 expression, whereas 65 patients exhibited high-level NUCB2 expression (Table 1). The chi-squared test was used to analyze the correlation between NUCB2 expression and clinicopathological parameters. The results revealed a significant association between NUCB2 expression and WHO grade (*p* < 0.0001). Kaplan–Meier analysis revealed that higher NUCB2 expression was significantly associated with worse survival outcomes in glioma patients (Figure 1B). Univariate analysis demonstrated that the WHO grade (*p* = 0.001) and NUCB2 expression (*p* = 0.001) were both associated with survival time (Table 2). Furthermore, multivariate analysis indicated that NUCB2 expression was independently associated with survival time (Table 2). These findings support the notion that NUCB2 is an independent prognostic factor in glioma.

### 3.2. Knockdown of NUCB2 Decreases Cell Viability, Migration, and Invasion Ability in GBM Cells

To compare NUCB2 expression between glial and GBM cells, the mRNA levels of NUCB2 were measured in the glial cell line (SVGp12) and various GBM cell lines (GBM8401, U87-MG, G5T/VGH, M059K, Hs683, and A172) using real-time polymerase chain reaction. All GBM cell lines exhibited higher levels of NUCB2 expression compared to the glial cell line (Figure 2). Subsequently, GBM8401 and U87-MG cells were transfected with NUCB2 siRNA (si-NUCB2#1 and si-NUCB2#2) or a nonsense siRNA (negative control group) to achieve NUCB2 knockdown. Western blotting was performed to confirm successful downregulation of NUCB2 protein expression in the si-NUCB2 groups, while no significant differences were observed between the control and negative control groups (Figure 3). To explore the role of NUCB2 in EMT, neovascularization, and cell proliferation, the protein levels of E-cadherin, N-cadherin, VEGF, and cyclin D1 were examined after NUCB2 knockdown in GBM8401 and U87-MG cells. Western blot analysis revealed that knockdown of NUCB2 led to increased E-cadherin levels and decreased levels of N-cadherin, VEGF, and cyclin D1 in both GBM8401 and U87-MG cells (Figure 3). Cell viability was assessed using the MTT assay, and the results demonstrated that NUCB2 siRNA#1 and NUCB2 siRNA#2 significantly reduced cell viability compared to the control and negative control groups on day 2 and day 3 in GBM8401 and U87-MG cells (Figure 4). Wound healing assays were performed to evaluate the effect of NUCB2 knockdown on migration ability. In both GBM8401 and U87-MG cells, migration ability was decreased following si-NUCB2#1 and si-NUCB2#2 treatments (Figure 5). The cell invasion capacity was evaluated through the Matrigel invasion assay, revealing that the suppression of NUCB2 in both the si-NUCB2#1 and si-NUCB2#2 groups led to a substantial reduction in the invasive potential of GBM8401 and U87-MG cells. (Figure 6).

### 3.3. NUCB2 Expression Increases Resistance of GBM Cells to TMZ

To examine the impact of NUCB2 expression on TMZ resistance, GBM8401 and U87-MG cells were transfected with NUCB2 shRNA or NUCB2 expression plasmid to induce knockdown or overexpression of NUCB2. The cells were then treated with TMZ, and cell viability was assessed using the MTT assay. The NUCB2 plasmid group exhibited significantly higher cell viability than the control group, while the NUCB2 shRNA group exhibited the lowest cell viability after TMZ treatment in both GBM8401 and U87-MG cells (Figure 7A). Western blot analysis revealed decreased levels of PARP and cleaved caspase-3 in the NUCB2 plasmid group, suggesting a decrease in apoptosis of tumor cells (Figure 7B).

### 3.4. NUCB2 Expression Enhances Radioresistance of GBM Cells via Increased DNA Repair Activities

To investigate the role of NUCB2 in radioresistance, GBM8401 and U87-MG cells transfected with shRNA or plasmid were treated with a single dose of 2Gy radiation. Colony formation assay was performed to evaluate cell survival. The results showed that the NUCB2 plasmid group had more colony formation than the control group, whereas the NUCB2 shRNA group exhibited fewer colonies (Figure 8A). After radiation, DNA damage occurs, and DNA repair activities are activated. Western blot analysis revealed that the NUCB2 plasmid group had higher protein levels of DNA repair-related proteins (Ku70, Ku80, Rad51, and Rad52) than the control group. Conversely, the NUCB2 shRNA group exhibited lower protein levels of these DNA repair proteins (Figure 8B).

### 3.5. Knockdown of NUCB2 Slows Intracranial Tumor Progression and Results in Longer Survival in the Animal Model

To investigate the impact of NUCB2 on tumorigenesis in vivo, an animal model with intracranial tumor implantation was used. GBM8401 cells transfected with luciferase gene and NUCB2 knockdown by shRNA were implanted into the brains of the animals. Tumor size was evaluated using IVIS imaging after tumor implantation. The level of luciferase detected in the NUCB2 knockdown group was significantly reduced compared to that in the control group (Figure 9A). On day 21 after tumor implantation, the intracranial tumors were removed, and immunohistochemical staining for NUCB2 expression was performed. The NUCB2 knockdown group showed less NUCB2 expression. Furthermore, the NUCB2 knockdown group exhibited significantly prolonged survival compared to the control group (Figure 9B).

## 4. Discussion

GBM is the most aggressive tumor in the central nervous system. The standard treatment is the Stupp protocol, established in 2006, which includes concurrent chemotherapy (TMZ) and radiotherapy. Complete tumor removal through surgery is challenging due to the tumor’s invasive nature into normal brain parenchyma. Surgery in eloquent brain areas can result in permanent neurological deficits. The use of tumor dyes during or prior to surgery can aid in distinguishing the tumor from normal brain tissue, although it is not at the cellular level. TMZ, an alkylating agent, methylates specific sites on DNA, primarily O^6^ of guanine, N3 of adenine, and N7 of guanine. Base excision repair can repair N^3^-methyladenine and N7-methylguanine, whereas O^6^-methylguanine is repaired by O6-methylguanine-DNA methyltransferase (MGMT). The expression of MGMT correlates with the treatment efficacy of TMZ. Radiotherapy induces tumor cell apoptosis by causing direct DNA damage or increasing reactive oxygen species. However, the resistance of GBM to TMZ can be as high as 50%, and the response to radiotherapy is often less than expected. This resistance may be attributed to metabolic alterations in GBM cells, the hypoxic tumor microenvironment, glioma stem cells, and the influence of miRNAs [25]. Moreover, NUCB2 knockdown does not affect MGMT expression, which suggests that NUCB2 might not be directly involved in the regulation of MGMT in GBM cells.

In our review, similar to other cancers, high NUCB2 expression correlates with higher pathological grading of glioma, and patients with high NUCB2 expression have a worse prognosis. GBM cell lines (GBM8401, GBM8901, U87-MG, and G5T) exhibit more prominent NUCB2 expression compared to the glial cell line (SVGp12). GBM8401 and U87-MG are both IDH wild-type GBM cell lines that were further studied in vitro. In GBM8401 cells, a decrease in NUCB2 expression was observed after treatment with siRNA (si-NUCB#1 and si-NUCB#2). Knockdown of NUCB2 expression by siRNA significantly decreased cell viability, migration ability, and invasion ability in GBM8401 cells. Intriguingly, our findings indicate that NUCB2 overexpression, in the absence of TMZ treatment, similarly influences cell growth as observed in the knockdown condition.

Neovascularization is a crucial step in cancer progression. In GBM, microvascular proliferation is one of the pathological criteria used for diagnosis. VEGF is a marker of neovascularization, and its expression is found in over 85% of patients with GBM. Anti-VEGF monoclonal antibodies can be used to treat recurrent GBM, but they have drawbacks such as systemic side effects and limited access to the central nervous system [26]. Knockdown of NUCB2 leads to a significant decrease in VEGF expression, suggesting that NUCB2 may be a potential target for regulating neovascularization in GBM.

Cyclins play a crucial role in cell cycle regulation. Cyclin D1 binds to cyclin-dependent kinase 4 and 6, activating the G1/S transition. High expression of Cyclin D1 is associated with the pathological grade and proliferative activity of astrocytomas [27,28]. Overexpression of Cyclin D1 induces TMZ resistance by upregulating P-gp in human malignant glioma cells. In our study, knockdown of NUCB2 downregulates Cyclin D1 expression in GBM8401 and U87-MG cells, suggesting that NUCB2 may be an upstream regulator of tumorigenesis and neovascularization. 

EMT is a process in which epithelial cells lose their apical-basal polarity and acquire a mesenchymal phenotype, enhancing their migration and invasion abilities. Transcription factors such as Snail, Slug, Zeb1, and Twist1 can induce EMT. During EMT, the expression of the epithelial marker E-cadherin decreases, whereas the expression of the mesenchymal markers N-cadherin and Vimentin increases. However, in GBM, the expression of N-cadherin may be similar in normal brain tissue and GBM. In specific subtypes of GBM, E-cadherin expression correlates with a worse prognosis [29,30,31,32]. Therefore, the classical E-to-N-cadherin switch observed in EMT may not be applicable to GBM and should be interpreted cautiously. Nevertheless, increased N-cadherin expression is associated with higher pathological grade, poor prognosis in glioma, and radioresistance in glioma stem cells. In our study, knockdown of NUCB2 by siRNA in GBM8401 and U87-MG cells resulted in increased E-cadherin levels and decreased N-cadherin levels. Further studies are needed to investigate whether the increase in N-cadherin induces a mesenchymal subtype of GBM.

To validate the knockdown effect of NUCB2 in vivo, a xenograft mouse model was established. GBM8401 cells with or without NUCB2 knockdown were implanted into the brains of athymic mice. The intracranial lesions were evaluated using an IVIS every three or four days until the 24th day after tumor implantation. The results demonstrate that knockdown of NUCB2 reduces tumor progression and prolongs survival in the mouse model. It is important to note that the intracranial growth inhibition observed in NUCB2 knockdown cells was expected due to the observed toxicity of NUCB2 knockdown in vitro, as indicated in Figure 4A. However, we did not experimentally address the specific impact of NUCB2 overexpression on intracranial tumor growth in this study. Future studies focusing on the effects of NUCB2 overexpression on in vivo tumor growth would provide valuable insights into its role in glioblastoma progression.

TMZ remains the only effective alkylating agent for treating GBM. However, more than 50% of patients show no response to TMZ, and almost all patients experience tumor recurrence eventually [5,33]. Several cellular alterations contribute to TMZ resistance, including enhanced DNA repair ability, modulation of autophagy, and epigenetic modifications [34]. Other factors, such as microRNAs, extracellular vesicles, and glioma stem cells, also play a role. Our data reveal that GBM8401 and U87-MG cells with NUCB2 knockdown exhibit significantly lower cell viability than the NUCB2 overexpression group when treated with different concentrations of TMZ. Increased apoptosis activity was observed in the NUCB2 knockdown group but not in the overexpression group after TMZ treatment in both GBM8401 and U87-MG cells, suggesting that NUCB2 overexpression leads to TMZ resistance in GBM cells.

Radiotherapy exerts its therapeutic effect by causing DNA breaks or increasing intracellular reactive oxygen species. DNA breaks can be repaired by DNA damage response mechanisms, or the irradiated cells may undergo cell death [35]. They are repaired by DSB repair mechanisms. The DDR is involved in the recognition of DSBs and regulation of repair. In GBM, there are hypoxic tumor regions where the tumor cells can better tolerate radiotherapy by modulating cellular processes, inhibiting apoptosis, regulating autophagy, and enhancing antioxidant ability. Additionally, tumor heterogeneity, metabolic alterations, and DNA repair pathway dysregulation can contribute to radiotherapy resistance in GBM [6]. In our study, knockdown of NUCB2 in GBM8401 and U87-MG cells resulted in reduced colony formation after radiation exposure, whereas cells overexpressing NUCB2 showed increased resistance to radiotherapy. 

There are two main DNA damage repair processes activated by double-strand breaks: non-homologous end joining (NHEJ) and homologous recombination (HR). Proteins Ku70 and Ku80 participate in NHEJ, whereas Rad51 and Rad52 are involved in HR [36,37]. The expression levels of these proteins were analyzed in GBM8401 and U87-MG cells after radiation treatment. The results revealed increased levels of Ku70, Ku80, Rad51, and Rad52 in the NUCB2 overexpression group, and decreased protein levels in the NUCB2 knockdown group in both GBM8401 and U87-MG cells. The increased radiation resistance in GBM cells is mediated by enhanced DNA repair abilities, while decreased expression of DNA repair proteins is associated with increased sensitivity to radiotherapy. It is worth noting that there are other DNA repair proteins, such as PARP1 (Poly ADP-ribose polymerase 1) and XRCC1 (X-ray repair cross-complementing protein 1), that are involved in DNA repair processes. However, it is important to clarify that this study did not investigate the expression or modulation of these specific proteins. Therefore, the exact mechanisms by which NUCB2 affects DNA repair in the context of radiotherapy would require further research.

Resistance to TMZ and radiotherapy is a complex interplay between tumor cells and the tumor microenvironment. In our study, knockdown of NUCB2 improved the sensitivity of GBM cells to TMZ and radiotherapy, whereas overexpression of NUCB2 enhanced resistance to treatment.

## 5. Conclusions

In conclusion, NUCB2 not only serves as a poor prognostic factor but also plays a significant role in the progression of GBM. Knockdown of NUCB2 inhibits tumor proliferation and neovascularization, and enhances the sensitivity of GBM to TMZ and radiation therapy. The mechanisms and pathways through which NUCB2 exerts its effects require further investigation. Nevertheless, NUCB2 represents a promising target in the treatment of GBM.

## Figures and Tables

**Figure 1 cells-12-02420-f001:**
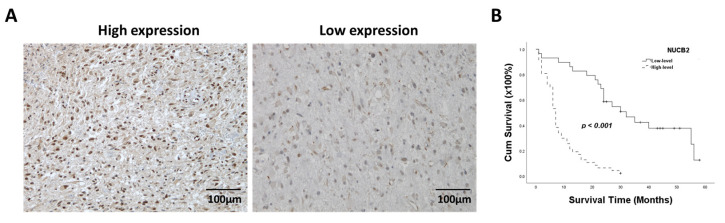
Correlation between NUCB2 protein levels and survival time in patients with glioma. (**A**) High and low levels of NUCB2 after IHC staining. (**B**) Kaplan–Meier analysis of NUCB2 expression.

**Figure 2 cells-12-02420-f002:**
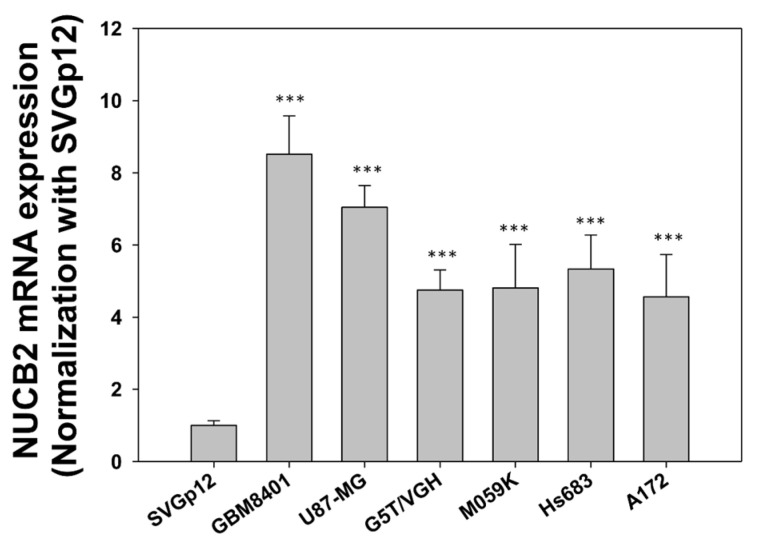
Real-time polymerase chain reaction was employed to analyze the expression levels of NUCB2 mRNA in both glial cells (SVGp12) and various types of GBM cells (GBM8401, U87-MG, G5T/VGH, M059K, Hs683, and A172). *** *p* < 0.001 compared with SVGp12 cells.

**Figure 3 cells-12-02420-f003:**
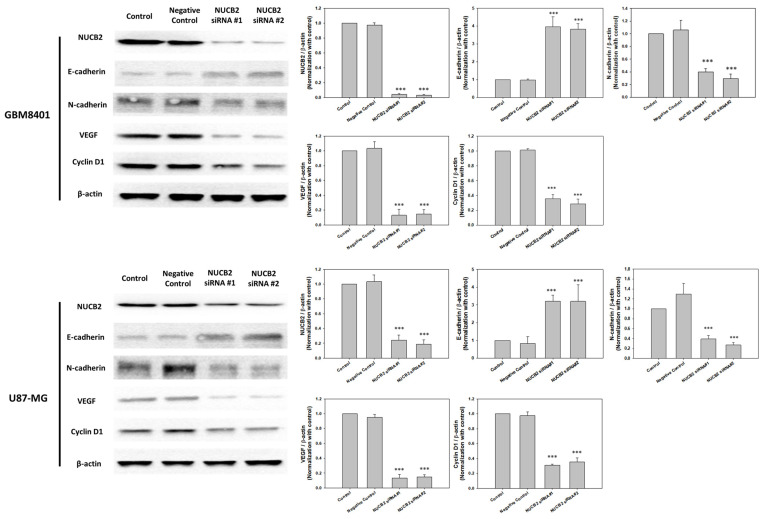
Western blot analysis was utilized to assess the expression levels of NUCB2, E-cadherin, N-cadherin, VEGF, and cyclin D1 in GBM8401 and U87-MG cells across four distinct groups: the control group, the negative control group, the si-NUCB2#1 group, and the si-NUCB2#2 group *** *p* < 0.001 compared with control group.

**Figure 4 cells-12-02420-f004:**
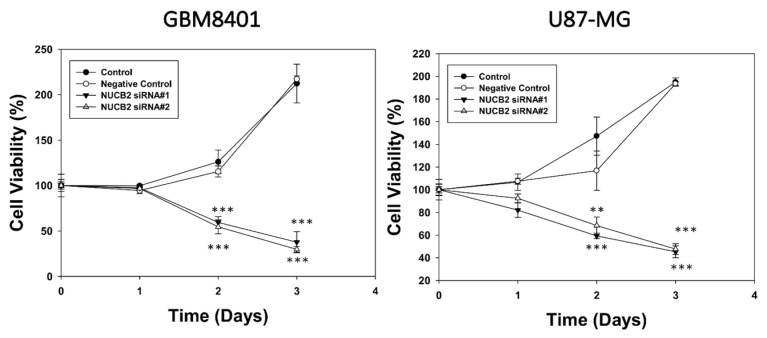
Cell viability was assessed using the MTT assay in GBM8401 and U87-MG cells across four groups: the control group, the negative control group, the si-NUCB2#1 group, and the si-NUCB2#2 group. ** *p* < 0.01 and *** *p* < 0.001 compared with control group.

**Figure 5 cells-12-02420-f005:**
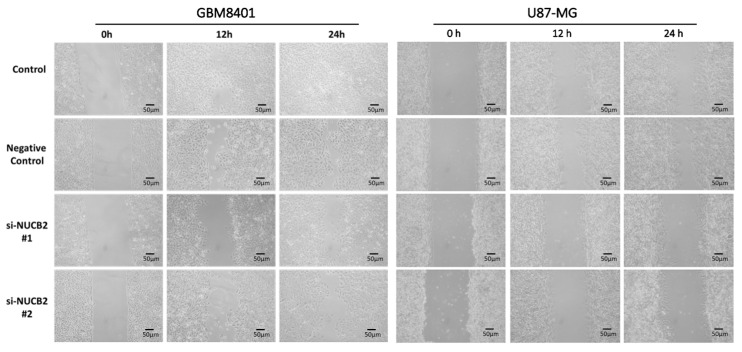
The cell migration capabilities were evaluated in GBM8401 and U87-MG cells using the wound healing assay across four groups: the control group, the negative control group, the si-NUCB2#1 group, and the si-NUCB2#2 group.

**Figure 6 cells-12-02420-f006:**
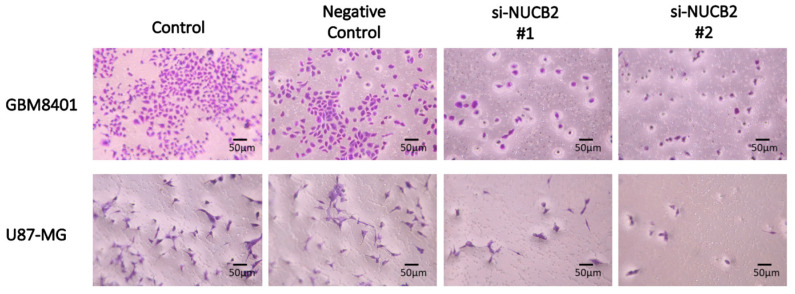
Cell invasion capabilities were assessed in GBM8401 and U87-MG cells utilizing the Matrigel invasion assay across four experimental groups: the control group, the negative control group, the si-NUCB2#1 group, and the si-NUCB2#2 group.

**Figure 7 cells-12-02420-f007:**
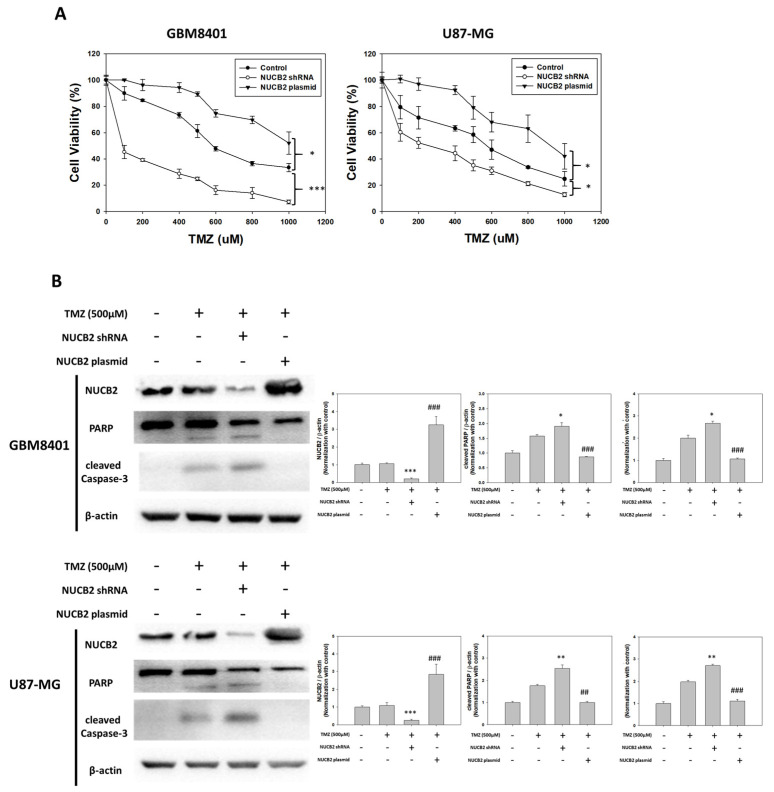
Effects of expression of NUCB2 on TMZ. (**A**) Cell viabilities of TMZ treatment combination between the control, NUCB2 shRNA, and NUCB2 plasmid groups in GBM8401 and U87-MG cells. (**B**) Protein levels of NUCB2, PARP, and cleaved caspase-3 as determined using Western blotting between the control, NUCB2 shRNA, and NUCB2 plasmid groups exposed to combined TMZ treatment in GBM8401 and U87-MG cells. * *p* < 0.05, ** *p* < 0.01 and *** *p* < 0.001 compared with TMZ only group. ## *p* < 0.01 and ### *p* < 0.001 compared with TMZ plus NUCB2 shRNA group.

**Figure 8 cells-12-02420-f008:**
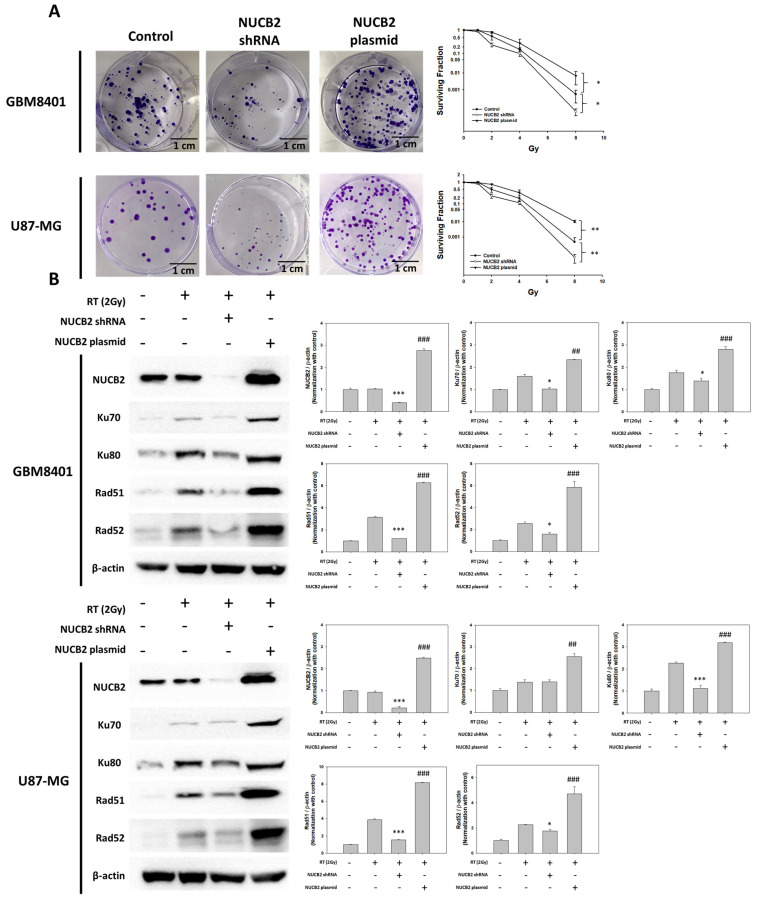
Therapeutic effects of NUCB2 combined with radiotherapy. (**A**) Colony formation with 2Gy radiotherapy between the control, NUCB2 shRNA, and NUCB2 plasmid groups. (**B**) The protein levels of NUCB2, Ku70, Ku80, Rad51, and Rad52 combined with radiotherapy as determined using Western blotting between the control, NUCB2 shRNA, and NUCB2 plasmid groups in GBM8401 and U87-MG cells. * *p* < 0.05, ** *p* < 0.01 and *** *p* < 0.001 compared with RT only group. ## *p* < 0.01 and ### *p* < 0.001 compared with RT plus NUCB2 shRNA group.

**Figure 9 cells-12-02420-f009:**
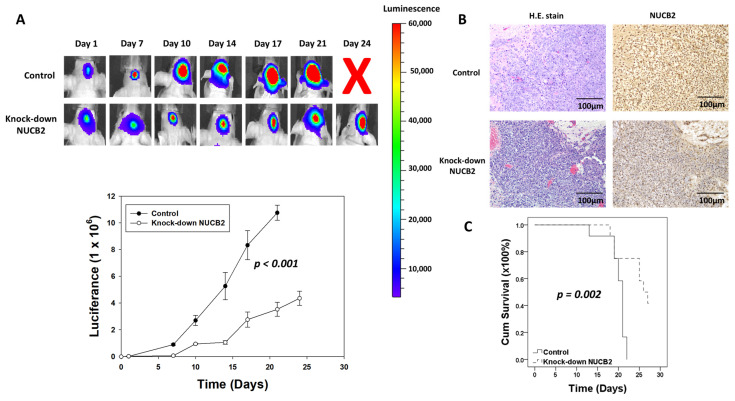
Animal model of GBM. (**A**) The tumor size was detected using luciferance between the control and knockdown NUCB2 groups. (**B**) The NUCB2 protein level following IHC staining on day 21 after tumor implantation between the control and knockdown NUCB2 groups. (**C**) The survival time following Kaplan–Meier analysis between the control and knockdown NUCB2 groups.

**Table 1 cells-12-02420-t001:** Correlation of NUCB2 expression with clinicopathologic parameters in patients with glioma.

	No. of Patients	NUCB2 Expression (*n*, %)		*p*-Value
		Low	High	
**Age (years)**				0.440
>60	25	7 (7.1%)	18 (18.2%)	
≤60	74	27 (27.3%)	47 (47.5%)	
**Sex**				0.847
Male	54	19 (19.2%)	35 (35.4%)	
Female	45	15 (15.2%)	30 (30.3%)	
**WHO Grade**				<0.0001 *
II	24	19 (19.2%)	5 (5.1%)	
III/IV	75	15 (15.2%)	60 (60.6%)	
**Tumor size**				0.372
≤3 cm	61	23 (23.2%)	38 (38.4%)	
>3 cm	38	11 (11.1%)	27 (27.3%)	
**Radiotherapy**				0.921
No	56	19 (19.2%)	37 (37.4%)	
Yes	43	15 (15.2%)	28 (28.3%)	
**Chemotherapy**				0.546
No	60	22 (22.2%)	38 (38.4%)	
Yes	39	12 (12.1%)	27 (27.3%)	
**KPS**				0.157
≤70	70	21 (21.2%)	49 (49.5%)	
>70	29	13 (13.1%)	16 (16.2%)	

* Statistical significance (*p* < 0.05). WHO, World Health Organization; KPS, Karnofsky performance score.

**Table 2 cells-12-02420-t002:** Univariate and multivariate analysis of different prognostic parameters without NUCB2 in patients with astrocytoma by Cox regression analysis.

	Univariate Analysis			Multivariate Analysis		
	Relative Risk	95% CI	*p*	Relative Risk	95% CI	*p*
**Age**	0.383	0.107–1.373	0.141			
**Gender**	0.552	0.151–2.015	0.368			
**WHO grade**	0.378	0.215–0.667	0.001 *	0.680	0.355–1.304	0.246
**Tumor size**	1.249	0.770–2.025	0.368			
**Chemotherapy**	1.000	0.630–1.586	0.999			
**Radiotherapy**	1.252	0.791–1.982	0.337			
**KPS**	1.540	0.926–2.564	0.096			
**NUCB2 expression**	0.178	0.098–0.324	0.001 *	0.210	0.109–0.403	<0.001 *

* Statistically significant (*p* < 0.05).

## Data Availability

No new data were created or analyzed in this study. Data sharing is not applicable to this article.

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
